# Coordination Investigation of the Economic, Social and Environmental Benefits of Urban Public Transport Infrastructure in 13 Cities, Jiangsu Province, China

**DOI:** 10.3390/ijerph17186809

**Published:** 2020-09-18

**Authors:** Xinghong He, Zhichao Cao, Silin Zhang, Shumin Liang, Yuyang Zhang, Tianbo Ji, Quan Shi

**Affiliations:** 1School of Transportation and Civil Engineering, Nantong University, Nantong 226000, China; 1826041030@stmail.ntu.edu.cn (X.H.); caozhichao@bjtu.edu.cn (Z.C.); 1826041038@stmail.ntu.edu.cn (S.L.); 1826041060@stmail.ntu.edu.cn (Y.Z.); tianbo.ji2@mail.dcu.ie (T.J.); sq@ntu.edu.cn (Q.S.); 2Chongqing Key Laboratory of Urban Rail Transit Vehicle System Integration and Control, Chongqing 400000, China

**Keywords:** public transport infrastructure benefit, coordination development, environment evaluation, investigation-based coordinated model, entropy method

## Abstract

This study proposed an investigation-based multiple-criteria coordinated model to evaluate the sustainable development of urban public transport (PT) infrastructure, based on economic, social and environmental data from 2009 to 2019. The main problem with the traditional approach for assessing urban PT development is that economic and social benefits are considered individually, but also attention to environmental factors and coordination among the three issues are nearly overlooked. This leads to the likelihood of inaccuracies in the handling/assessment of sustainable development or an imbalance among the attributes in different cities. An investigation-based coordinated model was introduced in which a survey of 35 sub-criteria was conducted to derive the criteria necessary for coupling/coordination. A case study involving 13 cities in Jiangsu Province, China, illustrated the problems in coordinating PT systems and verified the efficacy of the proposed approach. With employing the entropy method, this study validated coordination of the PT infrastructure development of various cities in a balanced manner and used panel regression formulas to analyse the theoretical gap and empirical bottlenecks existing among economic, social and environmental benefits. With the findings of the study, the data-based investigation from 13 cities enabled the city planners/managers (including ones from other cities with similar urban levels) to give the individual priority between the ternary benefits, advance technology, allow big data-based informatisation and implement near-future autonomous PT vehicles.

## 1. Introduction

The rapid development of urban public transport (PT) infrastructure enables human beings to travel in a more convenient manner [1,2], but also, from the perspective of sustainability, to promote environmental and economic redevelopment. However, heterogeneity and even unbalanced levels of development among different urban PT infrastructures is usually derived from capital accumulation and passenger demand. The geographical environment of various cities and the different political or economic status of these cities might lead to serious imbalances among the economic, social and environmental issues involved in the development of urban PT infrastructure. The “geographical environment” includes various PT modes, transport development directions and travellers’ behaviours of different urban terrains, such as flatlands and coastal cities. The main problem within these imbalances, from the perspective of urban sustainability, is that they frustrated new demand for PTs and retard progress in urbanisation and public health improvements [3,4,5]. The present work involves the development of an investigation-based coordinated model for estimating the criteria affecting sustainable development of urban public transport (PT) infrastructures, and the model is developed by considering economic, social and environmental data from the past decade. It should be noted that until the recent ten years (2009–2019), coordinated development of the 13 cities covering intensive development levels is less studied. Indeed, the modelling and findings are intended to be the reference for future development or other similar cites.

The development of urban PT infrastructure is vital for economic development [6], and it creates a development space for economic activities and sustains the flow of capital. Many scholars have studied the relationship between transportation and the economy [7,8]. Agbelie conducted an in-depth study of the work of three econometric frameworks for economy and transportation and concluded that specific economic sectors of each country depend significantly on transportation [7]. A considerable investment in PT construction is required to meet the needs for economic development. Indeed, PT infrastructure is built quickly. However, investment in PT infrastructure strictly aimed at economic benefits is overlooking the other dimensions of the development, namely social and environmental impacts. Imbalanced capital issue is the main obstacle and limiting factor for long-term economic development [8]. Hence, economic benefits have frequently been discussed in earlier evaluations of the benefits PT systems, but the coordination between social benefits and environmental benefits lacks the experience of modelling and empirical study.

The level of urban economic development in several developing countries is inconsistent, and the benefits of PT infrastructure at different stages of economic development make different contributions to economic, social and environmental benefits. It is not clear whether there is a significant relationship among the three benefits, and thereby quantitative data are still needed. Hereby, this study analysed the causal relationships existing among the economic, social and environmental benefits of PT infrastructure development and the contribution made to the economic environments of developing countries. As demonstrated in Figure 1, data were collected for the 13 sample cities Suzhou, Nanjing, Wuxi, Nantong, Changzhou, Xuzhou, Yangzhou, Yancheng, Taizhou, Zhenjiang, Huaian, Lianyungang and Suqian, all located in Jiangsu Province, China, and 18 evaluation indices were established. (i) The entropy weight method and the investigation-based coordination model were used to calculate the values of the three indicators, concluding the economic, social and environmental benefits. (ii) The internal connections between the benefits were explored with the panel regression model. (iii) The calculated results were analysed. It should be noted that the final public transportation infrastructure benefit is our evaluation result, as the so-called fourth benefit, which is derived from the three indicators with individual weightings: economic, social and environmental benefits.

The remainder of the paper is organised as follows. The article reviews previous studies and related literature in Section 2. The methodology is then introduced in Section 3. The results of the calculation are presented in Section 4. Subsequently, the experimental results are summarised and relevant policy implications are discussed in Section 5.

## 2. Literature Review

The economic issues relating to PT infrastructure were discussed extensively in previous studies. The relationship is primarily between the construction of PT infrastructure and economic development [9,10]. Transport infrastructure spending can increase productivity and economic output because of lower transport costs, improved market access and raw materials, shorter travel times, less traffic congestion and many other benefits [7,11]. The economic benefits of PT infrastructure are conducive to promoting urbanisation because convenient transportation makes commuting between rural and urban areas easier and more sustainable [12,13,14], promotes the growth of employment rate [15], probably leads to increases in housing prices and rents and thereby promotes the flow of capital [16,17], increases the profits of industrial enterprises [18] and even benefits the construction and development of cities and towns along the route, speeding up the process of urban communication [19,20]. There is a positive correlation between the growth of tourism output and the construction of high-speed railway network [21]. The construction of transportation infrastructure also makes the Asia Pacific region accessible to both Americans and Europeans, making the region a prosperous area for tourism [22]. Transportation infrastructure has become an important determinant of tourism inflow to destinations [23].

The social benefits of PT infrastructure include improvements in the living standards of urban residents and increases in the enrolment rates for higher education institutions [24,25]. The development of new industries is beneficial for recruiting new workers and raising urban employment rates [20,26], and the increased demand for labour raises the wage levels for employees and is helpful in increasing the disposable income of the urban population [27]. The construction of transportation facilities can provide the necessary conditions for local economic development and commercial prosperity, as well as increases the income level of the people along the route [28]. Additionally, urban PT infrastructure has a positive impact on the health of urban inhabitants [29,30].

Extensive research on the environmental benefits of PT infrastructure has been conducted to analyse environmental impact and particularly the sustainable development of cities [31]. The use of the PT infrastructure is also related to CO_2_ emissions [32], the long-term use of infrastructure construction can shorten the recovery time of greenhouse gases [33], and proper planning of a PT system helps to reduce greenhouse gas emissions [34]. Because the transportation sector is one of the main drivers of carbon emissions [35], it is important to determine the factors that affect carbon dioxide emissions in the transportation sector for the construction of low-carbon cities [36]. In addition, an excellent urban PT system encourages urban residents to choose it as their main mode of travel and reduces their reliance on private transportation. This reduces pollution and harmful gas emissions and improves the air quality in cities [37]. The reductions in the use of private transportation will also help to reduce traffic noise, thereby enhancing the quality of urban living and the working environment [38,39]. Maparu and Mazumder focussed on the short-term causal relationships existing among different subsectors of transport infrastructure and considered the direction of causality and the impacts on economic development and urbanisation [28]. Sun and Cui encouraged coordinated development by using a coupling coordination degree model, and they analysed the influence of economic, social and environmental criteria with a regression approach [40].

Existing studies show that the use of/investment in urban PT infrastructure has an evidently positive impact on the economic, social and environmental aspects of urban development. Stimulating economic development, promoting social progress and protecting the natural environment have always been among the primary benefits of urban PT infrastructure. Therefore, stimulating economic development, promoting social progress and protecting natural environment have been the basic benefits of urban public transportation infrastructure. However, variation rules or influence mechanism in the proportions of the economic, social and environmental benefits of urban PT within different economic circumstances have not yet been considered in quantitative analysis. Indeed, it is unknown whether there is a long-term quantitative causal relationship operating among these three benefits. Concerning this problem, this paper proposes an investigation-based multiple-criteria coordinated model to evaluate the sustainable development of urban public transport (PT) infrastructure, based on the historical/investigative data of economic, social and environmental in realistic case from 2009 to 2019.

## 3. Methodology

### 3.1. Economic, Social and Environmental Benefits of Urban PT Infrastructure

This study explicitly found the definitions of 3-tuple, in which: (i) economic benefit of urban PT infrastructure refers to the positive impact of construction of transportation facilities on economic development [9,10]; (ii) the social benefit of urban PT infrastructure is said to be the positive impact of construction of transportation facilities on social progress [12,13,14]; and (iii) the environmental benefit of urban PT infrastructure represents the positive impact of reducing private car trips because of construction of transportation facilities and includes the improvements to the natural environment. The 3-tuple benefits constitute the benefits of urban PT infrastructure jointly, contributing differently [31,32].

The three benefits of urban PT infrastructure also have certain interaction influences. An increase in the economic benefits of PT infrastructure is related to an augment in social benefits. The construction of the PT infrastructure creates a convenient transportation environment, promotes population flow and increases the rate of urbanisation [14,28]. The increase in the urbanisation rate is conducive to the expansion of an urban consumer market and to improvements in the supply of retail goods [24]. It promotes the prosperity of the trade economy and the entry of foreign enterprises and creates an employment gap; a growth in the number of employees increases the wage level of residents [26,27]. The relationship between competition and cooperation between foreign enterprises and local enterprises promotes the exchange and development of science and technology, and further improves the efficiency of production and the efficiency of resource allocation [20]. Furthermore, technological progress will facilitate faster economic development [8,31]. The rise in the basic environmental benefits of PT is also related to scientific and technological progress [33]. The advancement of technology is conducive to the development of efficient transportation with low carbon emissions, attracting more passengers, reducing the use of private transportation [39], reducing greenhouse gas emissions and improving the urban environment. Naturally, coordination evaluation becomes an important indicator for measuring the healthy development of a city [37]. It is well known that large-scale PT infrastructure is expensive, and the growth in economic benefits due to PT infrastructure is conducive to accelerating the rate of return. Ensuring investment in PT infrastructure is the basis for improving the scientific and technological level of transportation and protecting the natural environment.

### 3.2. Comprehensive Evaluation Model

Following Sun and Cui [40], Table 1 indicates the 18 related index evaluation systems proposed herein to calculate the benefits and extent of development coordination for PT infrastructure in the sample cities.

***Observation.*** The required data came primarily from the “2009–2019 China Statistical Yearbook”, “2009–2019 Yearbook”, “2009–2019 Statistical Development Bulletin” and “2009–2019 Environmental Statistics Bulletin” for each city. These data form the basis, and the rate of change for each indicator was calculated. Some data that are not released to the public and refer to the before and after values were calculated by using grey predictions and linear regression simulations. It should be noted that the readers can refer to the original data [41]. The predicted data are marked with a specific colour (grey).

The weights of the 18 evaluation indicators were calculated by using the entropy method. This study measured the degree of uncertainty by the means of entropy. Generally, the more information there is, the lower/smaller the uncertainty and entropy become. In term of the property of entropy, the degree of uncertainty of an event can be determined by computing the entropy value, or the dispersion degree of an indicator can be judged by the entropy value. The dispersion degree of the indicator is positively correlated with the influence of the indicator on the coordination evaluation. Hence, with respect to the degree of variation of each index, the information entropy can be used to obtain the weight of each index and derive a basis for the coordination evaluation of multiple indices. Generally, the classical coordination evaluation modelling framework [40,42] is applied to estimate multiple indices by suitable adjusted. Furthermore, how to transfer the empirical items as estimation indicators is one of the contributions in our study. The detailed procedures of the coordinated evaluation model, by developing the concept in [40,42], are as follows:

*Step 1.* Normalise processing and eliminate the impact of different indicator dimensions:(1)$$Y_{ij}=\frac{X_{ij}-min\left\{X_{j}\right\}}{max\left\{X_{j}\right\}-min\left\{X_{j}\right\}}$$
(2)$$Y_{ij}=\frac{max\left\{X_{j}\right\}-X_{ij}}{max\left\{X_{j}\right\}-min\left\{X_{j}\right\}}$$
where *X_ij_* represents the growth rate of the *j*th index in year *i* and *Y_ij_* represents the normalised data of the *j*th index in the *i*th year. When the decreasing value of indicator raises the level of urban public transportation infrastructure benefit, Equation (2) is captured. Otherwise, Equation (1) is used only if the increasing value of indicator enables improve the level of urban public transportation infrastructure benefit.

*Step 2.* Calculate the proportion of the *j*th index in the *i*th year:(3)$$\varpi_{ij}=\frac{Y_{ij}}{\sum_{i=1}^{m}Y_{ij}}$$
where *ϖ_ij_* represents the proportion of the *j*th indicator value in the *i*th year.

*Step 3*. Calculate the information entropy and redundancy of the *j*th index in the *i*th year:(4)$$e_{j}=-\frac{1}{Inm}\sum_{i=1}^{m}\left(\varpi_{ij}\times In\varpi_{ij}\right)$$(5)$$d_{j}=1-e_{j}$$
where *m* is the number of evaluation indicators, *e_j_* is defined as the information entropy of the *j*th indicator (*0 ≤ e_j_ ≤1*) and *d_j_* is the degree of redundancy in the entropy.

*Step 4.* Calculate weight *w_j_*:(6)$$w_{j}=\frac{d_{j}}{\sum_{j=1}^{m}d_{j}}$$
where *w_j_* is the weight of the *j*th indicator.

*Step 5.* Calculate weight *w**_l_*:(7)$$w_{l}=\sum_{j=1}^{p}w_{j}$$
where *w**_l_* is the weight of the *l*th benefit level and *p* is the number of indicators for the current benefit level.

*Step 6.* Calculate the benefit index for the urban PT:(8)$$S_{benefit,i}=\sum_{i=1}^{n}W_{j}Y_{ij}$$
where *S_benefit,i_* is the urban public transportation infrastructure benefit in the *i*th year.

*Step 7.* Calculate the three benefits of urban PT infrastructure:(9)$$Uli=\sum_{j=1}^{p}w_{j}Y_{ij}$$
where *U_li_* represents the *l*th benefit level in the *i*th year.

### 3.3. Coupling Coordination Model

The concept of coupling used in this study was originally derived from the field of physics. Herein, coupling is redefined as the dynamic relationship between two or more systems that affect each other. The degree of coupling reflects the extent of connection between the systems. The greater are the connections between systems, the bigger is the value of coupling. There are many interactions among the three benefits of urban PT infrastructure and the degree of their coupling affect the overall level of the PT infrastructure benefits of the city. However, in some cases, the degree of coupling does not account for the synergy between systems, and the degree of coordination is used to reflect this synergy [42]. Therefore, this study constructed a coupling coordination degree model to evaluate the degree of coordinated development among the three benefits of urban PT systems. The coupling coordination degree model is expressed as follows:(10)$$C={\left(\frac{U_{1}\times U_{2}\times U_{3}}{U_{1}+U_{2}+U_{3}}\right)}^{\frac{1}{3}}$$
(11)$$H=\sqrt{C\times S}$$
where *H* is the degree of coupling coordination and *C* is the degree of coupling between the three benefits. This study produced three benefits deriving from urban PT infrastructure: *U_1_*, *U_2_* and *U_3_* represent economic, social and environmental benefits, respectively. *H* can be used to illustrate the degree of coupling and coordination. This classification was based on the comparative relationship among three benefits [43]. The discriminating criteria based on [40] is shown in Table 2.

### 3.4. Panel Regression Model

The degree of coupling and coordination among the three benefits illustrates the level of coordinated development. When one of the three benefits fluctuates, it causes the other two benefits to change, and the coordination among them and even the overall benefits of the urban PT infrastructure also change accordingly. To analyse the impacts of the three benefits on the overall benefits derived from the urban PT infrastructure, an empirical model was established. Because the sample set contains data for different cities, this study uses a panel regression model to analyse the impact of the three types of benefits, the benefits of urban PT infrastructure, and the extent of the impact between two benefits. The following panel regression model formulas are presented.
(12)$$S(i,t)=\alpha_{0}+\alpha_{1}U_{l}(i,t)+\alpha_{2}U_{l}(i,t)+\alpha_{3}U_{l}(i,t)+\beta$$
(13)$$U_{l}(i,t)=\lambda_{0}+\lambda_{1}U_{l}(i,t)+\lambda_{2}U_{l}(i,t)+\delta$$
where *S(i,t)* are the PT infrastructure benefits of city *i* in year *t* and *U*_1_
*(i,t)*, *U*_2_
*(i,t)* and *U*_3_
*(i,t)* represent the economic, social and environmental benefits resulting from the urban PT system, respectively. The subscript of *U*, *l*, represents a variable with the range of 1–3. For the structure of city *i* in year *t*, $\alpha_{0}$ represents a constant term and parameters *α*_1_, *α*_2_ and *α*_3_ represent the elasticity estimates of *S* with respect to *U*_1_, *U*_2_ and *U*_3_, respectively. *λ*_0_ is a constant term that is the value in the polynomial due to the panel regression, in a sum that does not multiply any expression involving the independent variable. The parameters *λ*_1_ and *λ*_2_ represent the elasticity estimates of *U*_l_. *β* and *δ* are also constant terms.

## 4. Results and Discussion

### 4.1. Impact of Urban PT Infrastructure

The individual benefit weighting factors of the 13 cities are shown in Table 3 based on the proposed entropy method. Figure 2 demonstrates economic, social and environmental benefits for the thirteen cities, calculated according to Equations (8) and (9) and the coupling coordination degree calculated according to Equation (11). For convenience, this paper has legends to express multi-criteria. That is, this paper uses “*U*_1_: economic”, “*U*_2_: social”, “*U*_3_: environmental”, “*S*: infrastructure benefits” and “H: coupling value” in Figure 2.

### 4.2. Economic Segment

The systematic clustering method considers the clustered samples or variables as a group, determines the similarity statistics between the grades, selects the closest two or more grades to incorporate into a new grade, calculates the new class and finally, for the similarity statistics among other categories, the closest two or several groups are selected and incorporated into a new category until all samples or variables are incorporated into one category.

After clustering, in Figure 3, the abscissa is the Euclidean distance and the ordinate is the index of 13 cites individually. Using the 10-year GDP data for the sample cities, SPSS software was employed for systematic clustering and the result was divided into the five grades denoted as T1–T5. T1 includes Suzhou; T2 includes Nanjing and Wuxi; T3 includes Nantong, Changzhou and Xuzhou; T4 includes Yangzhou, Yancheng, Taizhou and Zhenjiang; and T5 includes Huaian, Lianyungang and Suqian.

From the data in Table 4 and the horizontal comparison, it can be seen that the economic benefits of PT infrastructure in cities of all levels are relatively significant, and there are multiple relationships existing between social and environmental benefits. In the longitudinal comparison, T1 cities exhibit the highest economic benefits and the lowest environmental benefits; T2 cities have the highest environmental benefits; T3 cities show the smallest difference among economic, social and environmental benefits; and T4 and T5 cities have the highest economic benefits and the lowest social benefits. The trend for environmental benefits is flat, but the level is not high. Generally, urban cities rely on PT in very different ways than rural cities. Herein, there is an effect emerging of “less developed” cities versus “more developed” cities having different needs from transportation. For example, the types of jobs comprising the economic base for the city, i.e., the requirements that drive transportation in the first place, can be different depending on the behaviours in that city.

As illustrated in Figure 4, the benefit characteristics of the PT infrastructure in the T1 cities are as follows: economic benefits are primary, social benefits are subordinate and environmental benefits are low; the coordination degree is 0.34; and the level is four. This corresponds to slightly unbalanced development. As part of the development of transportation resources, the length of the PT infrastructure road network increased between 2009 and 2019. At the same time, the GDP of Suzhou has also increased annually. Therefore, the weight of the economic benefit within the T1 urban PT infrastructure benefits is relatively large. With the exception of environmental benefits, other benefits increased slowly from 2009 to 2011. In 2012, however, Suzhou authorities began operating a night bus. In 2013, five transportation companies in Suzhou consolidated resources and merged into a single PT company. Since then, the basic, economic and social benefits of urban PT have increased rapidly, reaching first place among the groups for the period 2009–2019. In 2016, Suzhou rail transit accounted for 16.3% of public passenger transport. By 2018, rail transit accounted for 33.2% of public passenger transport, a level 2.03 times that of 2016. Rail transit is obviously developing rapidly. The basic benefit of PT was 0.3250, the economic benefit was 0.4523 and the social benefit was 0.2392 and rising rapidly, reaching the second rank for 2009–2019 and exhibiting the greatest growth over the ten years.

The benefits of PT infrastructure in the T2 cities are characterised in Figure 5. Economic benefits are the mainstay; social benefits and environmental benefits are approximately equal and their levels are low. The changes in social benefits are gradual, the environmental benefits vary widely and the overall coordination degree is 0.1249. In the second level, there is seriously unbalanced development and it is at a low level. Nanjing authorities opened Metro Line 1 in 2005 and Metro Line 2 in 2010. Between 2010 and 2011, the basic benefits of urban PT increased, and the environmental benefits have risen sharply. It is inferred that the use of rail transit has reduced the number of private car trips. As a result of reducing carbon emissions and greenhouse gas emissions, the air quality rate has improved. In 2014, Wuxi Metro (Lines 1 and 2) was put into operation, Nanjing opened the first tram in Jiangsu Province and Metro Line 10, S8, in the same year. The economic benefits of urban PT infrastructure in the T2 cities have reached their peak, but the environmental benefits have dropped significantly. The reason for this is that the convenience of transportation has greatly promoted economic development, corporate profits have risen, employment rates have risen, per capita disposable income has increased and private car travel has increased, and this has greatly reduced environmental benefits. In 2015, Nanjing Metro Line 3 was put into operation, with a total length of 44.9 km. In early 2017, Metro Line 4 was put into operation. At the end of 2017, Nanjing opened the Kirin tram. Wuxi’s GDP, fiscal revenue, tourism economy and industrial profits have risen sharply. At this time, the economic benefits of PT infrastructure have reached a ten-year maximum.

As illustrated in Figure 6, the benefits of PT infrastructure in T3 cities include economic impact as primary benefit, followed by social benefits and then environmental benefits, and the overall coordination degree is 0.2266. Development is moderately unbalanced, and the general trend for PT infrastructure exhibits the greatest impact from economic benefits, moderate impact from social benefits and weaker impact from environmental benefits.

The benefits of PT infrastructure in T4 cities rank are given in Figure 7. Economic benefits are first, followed by social benefits and then environmental benefits, with an overall coordination degree of 0.1929, and development is seriously unbalanced. The general trend for PT infrastructure benefits shows that the maximum impact comes from economic benefits and that this impact differs greatly from that of social benefits, while environmental benefits have not changed much in the past decade. The analysis shows that rail transit in the T4 cities is still in its infant stage, and PT has not provided an alternative to private travel. Therefore, environmental benefits have developed slowly. The transportation policies of T4 cities focus primarily on economic benefits, and there is still room for improvement in the economy without considering sustainable development and coordinated development.

The benefits of PT infrastructure in T5 cities are shown in Figure 8. Economic benefits are greatest, social benefits and environmental benefits are approximately equal and the overall coordination degree is 0.2601; the level is three, corresponding to moderately unbalanced development. In general, the trend in PT infrastructure benefits runs closest to that of economic benefits, roughly similar to social benefits, and slightly closer to environmental benefits, and the level of coordinated development is acceptable.

Based on the empirical investigation above, it is possible that the spikes and dips in certain years are derived from various metro lines put into operation, as in Nanjing and Wuxi cities. The fluctuations in most cites have similar reasons of new construction. Hence, from the viewpoint of the study, it is normal for these fluctuations to occur. In addition, these investments indeed include just new construction.

From the above five benefit curves (Figure 5, Figure 6, Figure 7 and Figure 8), the benefits of urban PT in the T1 grade are identical to economic benefits and social benefits, whereas the cities in the T2 grade do not reflect the characteristics described above. However, cities in the T3–T5 grades have changed significantly, and the basic and economic benefits of urban PT are basically the same. With increases in Tl grade, this feature becomes increasingly obvious.

From the perspective of coordinated development, all coordination degrees for all levels of cities are below 0.4; the highest average is 0.34 for T1, while the lowest is 0.1249 for T2. The values for T3 and T5 are relatively close at 0.2266 and 0.2601, respectively, and the value for T4 is 0.1929. The city with the highest economic level has the highest degree of coordination. The city with the second highest economic level has the lowest degree of coordination. In the benefit structure for this level of city, the proportion of environmental benefits is greater than 0.4. It is assumed that this value is due to the lagging coordination realised during this transitional period. The degree of coordination between T3 and T5 cities is similar, and the benefit structures for these cities give highest priority to economic benefits while social and environmental benefits are accorded basically equal emphasis. The common feature of the PT system is that it has not yet been developed to include rail transportation and the advantage of leading economic benefits has not realised.

### 4.3. Five-Echelon Panel Returns

#### Significance of the Three Benefits to the Basic Benefits of Urban PT

Table 5 denotes the definition of internal causality between benefits. The panel regression of the 13 city samples shows that the economic, social and environmental impacts on PT infrastructure are all significant and positive, and the regression coefficients are 0.527, 0.259 and 0.202, respectively. Economic benefits and social benefits have a two-way causal relationship, and there is no pairwise relationship between other pairs of benefits.

Panel regression was performed on samples of T5 grade cities and it is concluded that the economic benefits of PT infrastructure are significant, with a positive impact and regression coefficient of 0.472, whereas social and environmental benefits are not significant. As with the previous result, economic and social benefits have a two-way causal relationship, and there is no pairwise relationship between other benefits.

Table 6 shows that U1 exhibits significance at the level of 0.01 (t = 23.307, *p* = 0.000 < 0.01) and the regression coefficient value is 0.527 > 0, indicating that U1 has a significant positive impact on the S relationship. U2 shows significance at the 0.01 level (t = 5.546, *p* = 0.000 < 0.01) and the regression coefficient value is 0.259 > 0, indicating that U2 have a significant positive influence on S. U3 shows significance at the level of 0.01 (t = 5.340, *p* = 0.000 < 0.01) and the regression coefficient value is 0.202 > 0, indicating that U3 has a significant positive influence on S.

Table 7 shows that U1 shows a significance level of 0.01 (t = 4.497, *p* = 0.000 < 0.01) and the regression coefficient value is 0.47 2 > 0, indicating that U1 has a significant positive impact on the S relationship. U2 did not show significance (t = 1.929, *p* = 0.060 > 0.05), indicating that U2 does not have an influence on S. U3 also does not show significance (t = 1.176, *p* = 0.246 > 0.05), indicating that U3 has no influence on S.

## 5. Conclusions

Three concepts can be drawn from the experimental data. First, the results show that the environmental benefits of urban PT infrastructure are scarcely dependent on economic and social benefits. Based on five indicators to assess environmental benefit, namely air quality; quantities of SO_2_, NO_2_ and particulate pollutants; and traffic noise, the variation in environmental benefit has no correlation impact on economic and social benefit. Moreover, they did not generate a causal relationship in a quantitative manner. In addition, the values of the environmental benefits for urban PT infrastructure have always been less fluctuated and in smaller order of magnitude compared with other two benefits.

The second viewpoint is that the economic and social benefits of urban PT infrastructure exhibit a two-way causal relationship, and both impart positive effects. From perspective of either a separate analysis of thirteen cities or a five-grade analysis, the increase in economic benefits from urban PT infrastructure is conducive to the increase in social benefits, and vice versa. Based on the above analysis, if city planners/managers are inclined to improve the social benefits resulting from urban PT infrastructure, it is reasonable to give the priority to enhance the economic benefits of urban PT infrastructure. This concept is used in the development of transportation policies of several cities, such as Suzhou, Yangzhou, Zhenjiang, Lianyungang and Taizhou. However, due to their two-way, positive causal relationship, if planners/managers intend to improve the economic benefits resulting from urban PT infrastructure, they can give the priority of applying big data-based information, autonomous PT vehicles implementation [1,2] and even 5G-derived Internet of things engineering in the near-future to make urban PT serviceability more attractive.

The third view is that, when exploring cities with large differences in total GDP, the benefits of PT infrastructure are solely related to economic benefits and have somewhat distant related with social and environmental benefits. When analysing cities with small differences in GDP, economic, social and environment benefits all exhibit a significant positive correlation with the benefits of PT infrastructure. The computational/graphical results demonstrate that, with developments in the economy, the roles of social and environmental benefits have gradually become more prominent. When substantial economic development of a city has yet to be realised, the benefits of PT infrastructure are mainly economic benefits. When the city’s GDP reaches a threshold, the benefits of PT infrastructure are composed of economic, social and environmental benefits, and the overall 3-tuple is important. In other words, when the GDP of a city reaches the threshold, urban development is not only based on the accumulation of capital, but also on the advancement of science and technology.

The data-based investigation introduced by this study has a realistic sphere of forecasts for PT development in different phases. Future development suggestions, some of which are derived from theoretical evidence on the basis of the coordinated model, include consideration of: (i) Cities with different levels of economic development intend to enact different transportation policies. For a developing country in which the economic level of an inner city is quite different from those of other cities, transportation policies can be formulated according to the economic differences between cities. With gaining the feasible /suitable experience of available transportation policy cases, one city should also refer to the relative GDP levels. (ii) During the recent COVID-19 pandemic, several PT systems were completely or partly suspended. As they are gradually resuming operations, economic benefits should be prioritised. The reverse is also true. If one is to improve the economy quickly after the epidemic, one must first ensure the operation management rule (one component of social benefit) of the PT system.

## Figures and Tables

**Figure 1 ijerph-17-06809-f001:**
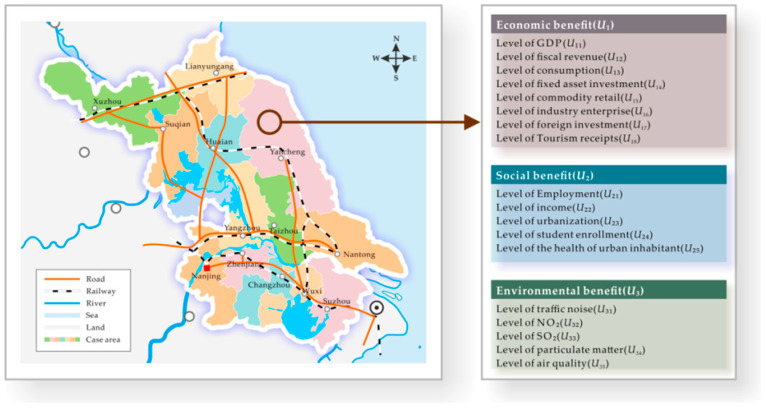
Map of 13 cities in Jiangsu Province of China.

**Figure 2 ijerph-17-06809-f002:**
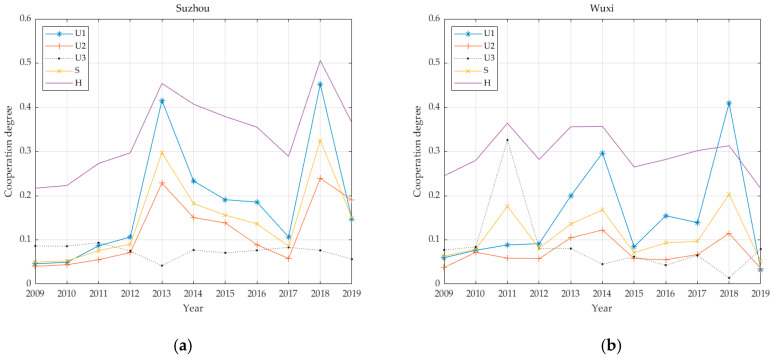
Thirteen cities 2009–2019 urban PT economic, social, environmental benefits and coupling coordination. (**a**) Suzhou; (**b**) Wuxi; (**c**) Nanjing; (**d**) Nantong; (**e**) Changzhou; (**f**) Xuzhou; (**g**) Zhenjiang; (**h**) Yancheng; (**i**) Taizhou; (**j**) Yangzhou; (**k**) Lianyungang; (**l**) Huaian; (**m**) Suqian.

**Figure 3 ijerph-17-06809-f003:**
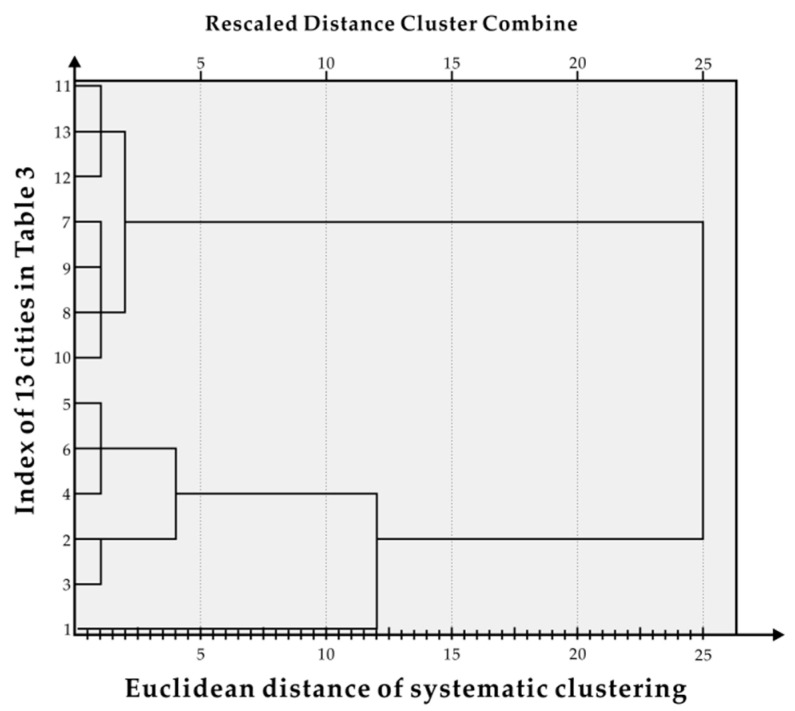
Hierarchical clustering results.

**Figure 4 ijerph-17-06809-f004:**
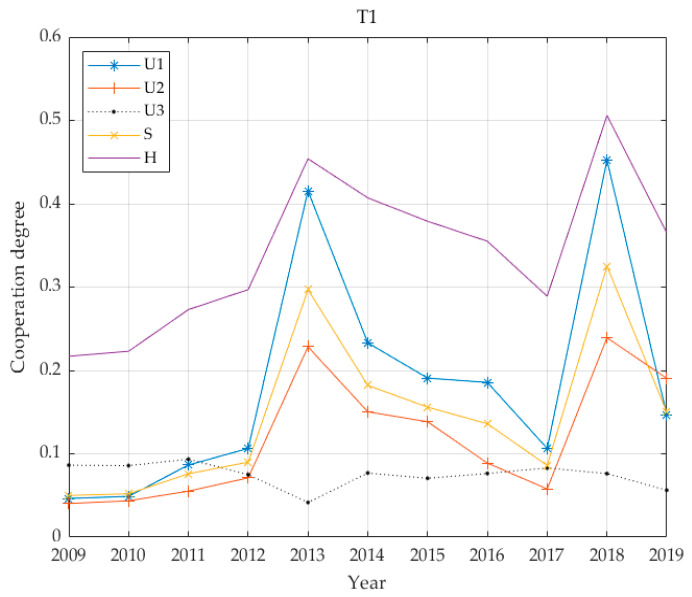
Four benefit values and coupling coordination degree of T1 grade.

**Figure 5 ijerph-17-06809-f005:**
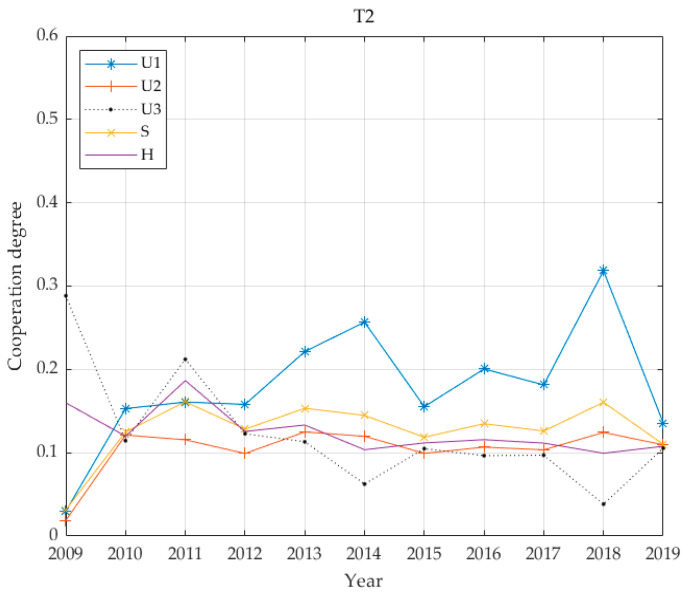
Four benefit values and coupling coordination degree of T2 grade.

**Figure 6 ijerph-17-06809-f006:**
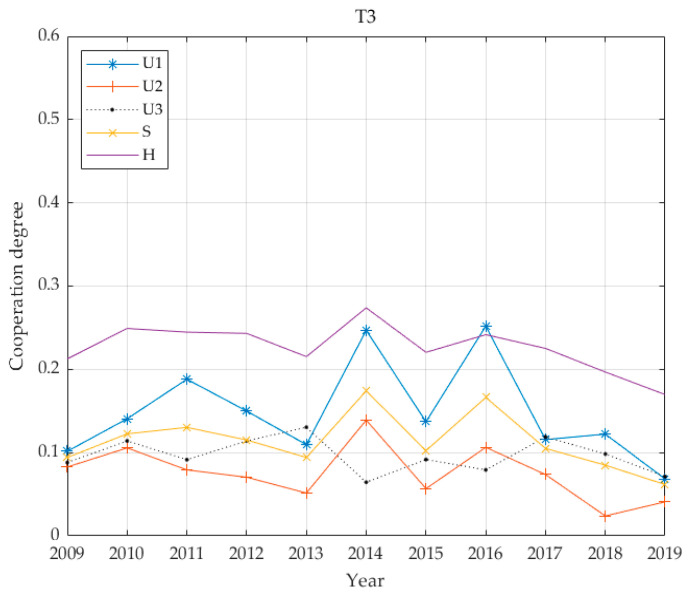
Four benefit values and coupling coordination degree of T3 grade.

**Figure 7 ijerph-17-06809-f007:**
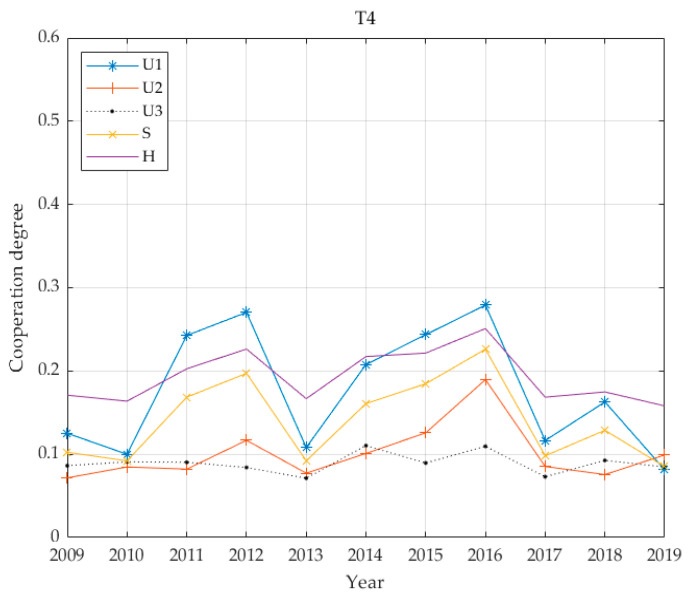
Four benefit values and coupling coordination degree of T4 grade.

**Figure 8 ijerph-17-06809-f008:**
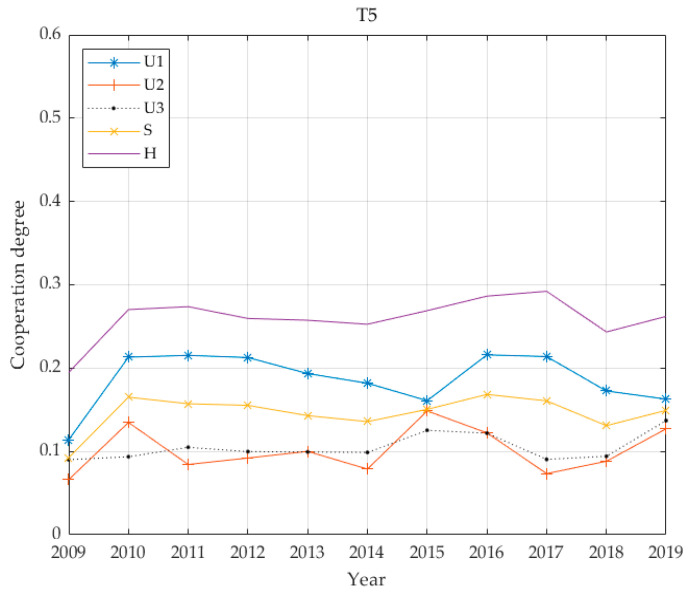
Four benefit values and coupling coordination degree of T5 grade.

**Table 1 ijerph-17-06809-t001:** Details of 18 evaluation indicators.

Level 1	Level 2	Description
Economic benefit (U1)	Level of GDP (U11)	(Change rate of GDP/Change rate of urban PT infrastructure level)*100%
	Level of fiscal revenue (U12)	(Change rate of fiscal revenue/ Change rate of urban PT infrastructure level)*100%
	Level of consumption (U13)	(Change rate of average consumption expenditure of urban dwellers/Change rate of urban PT infrastructure level)*100%
	Level of fixed asset (U14)	(Change rate of amount of fixed investment asset investment/Change rate of urban PT infrastructure level)*100%
	Level of commodity retail (U15)	(Change rate of total retail amount of commodity/Change rate of urban PT infrastructure level)*100%
	Level of industry enterprise profit (U16)	(Change rate of total profit of industry enterprise/Change rate of urban PT infrastructure level)*100%
	Level of foreign investment (U17)	(Change rate of actual utilised foreign investment/Change rate of urban PT infrastructure level)*100%
	Level of Tourism receipts (U18)	(Change rate of tourism receipts/ Change rate of urban PT infrastructure level)*100%
Social benefit (U2)	Level of Employment (U21)	(Change rate of employed population/Change rate of urban PT infrastructure level)*100%
	Level of income (U22)	(Change rate of disposable income Per capita /Change rate of urban PT infrastructure level)*100%
	Level of urbanisation (U23)	(Change rate of urbanisation rate/ Change rate of urban PT infrastructure level)*100%
	Level of student enrolment (U24)	(Change rate of student enrolment of regular higher education school/Change rate of urban PT infrastructure level)*100%
	Level of the health of urban inhabitant (U25)	(Change rate of death rate of population/Change rate of urban PT infrastructure level)*100%
Environmental benefit (U3)	Level of traffic noise (U31)	(Change rate of average value of traffic noise/Change rate of urban PT infrastructure level)*100%
	Level of NO_2_ (U32)	(Change rate of average daily NO_2_/Change rate of urban PT infrastructure level)*100%
	Level of SO_2_ (U33)	(Change rate of average daily SO_2_/Change rate of urban PT infrastructure level)*100%
	Level of particulate matter (U34)	(Change rate of average daily particulate matter/Change rate of urban PT infrastructure level)*100%
	Level of air quality (U35)	(Change rate of excellent rate of air ambient quality/Change rate of urban PT infrastructure level)*100%

Note: GDP is the abbreviation of Gross Domestic Product.

**Table 2 ijerph-17-06809-t002:** Coupling coordination interval.

H	Class
0.000–0.001	Extremely unbalanced development
0.101–0.200	Seriously unbalanced development
0.201–0.300	Moderately unbalanced development
0.301–0.400	Slightly unbalanced development
0.401–0.500	Barely unbalanced development
0.501–0.600	Barely balanced development
0.601–0.700	Slightly balanced development
0.701–0.800	Moderately balanced development
0.801–0.900	Favourably balanced development
0.901–1.000	Superiorly balanced development

**Table 3 ijerph-17-06809-t003:** Three benefit weighting factors of 13 cities.

Index	City	Weighting Factors (2009–2019)		
		Economy	Society	Environment
1	Suzhou	0.5064	0.3566	0.1369
2	Wuxi	0.4336	0.1777	0.3888
3	Nanjing	0.2568	0.1842	0.5589
4	Nantong	0.3430	0.4358	0.2212
5	Changzhou	0.4820	0.2663	0.2517
6	Xuzhou	0.4804	0.3129	0.2067
7	Zhenjiang	0.5630	0.2940	0.1430
8	Yancheng	0.4804	0.2812	0.2384
9	Taizhou	0.5771	0.2701	0.1528
10	Yangzhou	0.5182	0.2676	0.2143
11	Lianyungang	0.5876	0.1796	0.2328
12	Huaian	0.3777	0.2787	0.3436
13	Suqian	0.4073	0.3834	0.2092

**Table 4 ijerph-17-06809-t004:** Five grades of weight.

	City	Weight (2009–2019)		
		Economy	Society	Environment
T1 More developed	Suzhou	0.5064	0.3566	0.1369
T2 Well developed	Nanjing	0.3452	0.1809	0.4738
	Wuxi			
T3 ordinary	Nantong	0.4351	0.3383	0.2265
	Xuzhou			
	Changzhou			
T4 Undeveloped	Yangzhou	0.5347	0.2782	0.1871
	Yancheng			
	Taizhou			
	Zhenjiang			
T5 poor	Huaian	0.4575	0.2806	0.2619
	Suqian			
	Lianyungang			

**Table 5 ijerph-17-06809-t005:** Definition of internal causality between benefits.

Contrast	Five-Grade Cities		Thirteen Cities	
	Direction	Relationship	Direction	Relationship
U1 and S	U1 to S	Unidirectional	U1 to S	Unidirectional
U2 and S	No causality		U2 to S	Unidirectional
U3 and S	No causality		U3 to S	Unidirectional
U1 and U2	Bidirectional		Bidirectional	
U1 and U3	No causality		No causality	
U2 and U3	No causality		No causality	

**Table 6 ijerph-17-06809-t006:** Panel data for 13 cities.

Thirteen Cities					
Item	Regression Coefficients	Standard Error	t (Test)	*p* Significance	95% CI (Confidence Interval)
Intercept	−0.001	0.006	−0.178	0.859	−0.012~0.010
U1	0.527	0.023	23.307	0.000 **	0.482~0.571
U2	0.259	0.047	5.546	0.000 **	0.168~0.351
U3	0.202	0.038	5.34	0.000 **	0.128~0.277
F (3,127) = 485.439, *p* = 0.000					
R² = 0.920, adjust R² = 0.910					

Note: “**” is the denotation of one percent significance level.

**Table 7 ijerph-17-06809-t007:** Panel data for five-grade cities.

Five-Grade Cities					
Item	Regression Coefficients	Standard Error	t (Test)	*p* Significance	95% CI (Confidence Interval)
Intercept	0.003	0.027	0.113	0.911	−0.050~0.056
U1	0.472	0.105	4.497	0.000 **	0.267~0.678
U2	0.389	0.201	1.929	0.06	−0.006~0.783
U3	0.203	0.172	1.176	0.246	−0.135~0.540
F (3,47) = 23.174, *p* = 0.000					
R² = 0.597, adjust R² = 0.537					

Notes: ** *p* < 0.01.

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
