# Peer review of "Coordination Investigation of the Economic, Social and Environmental Benefits of Urban Public Transport Infrastructure in 13 Cities, Jiangsu Province, China"

_ijerph, 2020, doi:10.3390/ijerph17186809_

Round 1

Reviewer 1 Report

Thank you for the opportunity of reviewing this interesting article.  My general comment is that the manuscript includes valuable findings, but I cannot recommend this paper for publication as is. However, I suggest the following to improve the quality of the paper

In Title should be pointed out that this is a case study.

The literature review is not extensive and exhaustive.

Please provide better Figures

Please use throughout the manuscript the third person 

In 3.1 is not given any reference

It should be pointed out if Equations 1-10 were developed by authors

Line 45: the model was developed by authors?

Line 72: of the four benefits?

Lines 168-169: please give a bibliography according to guidelines of the journal

Line 225: Table 2 is based on which research?

Line 236-237: Pleas define β  and  δ in formulas 11-12     

Section 4.1: Please give the process used for calculating Weighting factors

Discussion section should critically discuss the findings/results of the case study in conjunction to findings of previously published case studies

Author Response

Response to reviewers and the editor- ijerph-911535, Sep.-2020

Coordination investigation of the economic, social, and environmental benefits of urban public transport infrastructure in 13 cities, Jiangsu Province, China

Xinghong He, Zhichao Cao, Silin Zhang*, Shumin Liang, Yuyang Zhang, Tianbo Ji, and Quan Shi

Note: All changes and revisions made in the revised manuscript are highlighted for convenience.

Reviewer: 1

Comments:

Thank you for the opportunity of reviewing this interesting article. My general comment is that the manuscript includes valuable findings, but I cannot recommend this paper for publication as is. However, I suggest the following to improve the quality of the paper.

Response: Your view making us to try harder to improve our paper. We hope that after our revisions and clarifications you’ll change your mind. We appreciated your good comments and followed to revise the paper, and hope that the revised version is clearer. All changes made are highlighted.

  1. In Title should be pointed out that this is a case study.

Response: You are right. Thanks. Changed (it is highlighted) in title.

  1. The literature review is not extensive and exhaustive.

Response: Following your comment, we added more related and necessary literature reviews (highlighted) in Section 2.

  1. Please provide better Figures.

Response: Thanks. Indeed, all the figures have been redraw already and improved much.

  1. Please use throughout the manuscript the third person. 

Response: Following your advice, we revised and ensured the third person uniformly.

  1. In 3.1 is not given any reference.

Response: Thanks again. Following your suggestion, we added relevant literature review in 3.1. It is also highlighted.

  1. It should be pointed out if Equations 1-10 were developed by authors.

Response: Please allow us to explain clearer. Generally speaking, the classical coordination evaluation modelling framework is applied to estimate multiple indexes by suitable adjusted. Furthermore, how to transfer the empirical items as estimation indicators is one of the contributions in our study. In addition, we added the references and the statement, highlighted on Lines 199-202.

  1. Line 45: the model was developed by authors?

Response: Thanks for your question that is similar to the above one. We added the necessary statement on Lines 199-202.

  1. Line 72: of the four benefits?

Response: We added the explanation on Lines 80-83 to make clearer. Thanks.

  1. Lines 168-169: please give a bibliography according to guidelines of the journal

Response: OK. Well done on Line 189. Thanks.

  1. Line 225: Table 2 is based on which research?

Response: We added the reference on Lines 249-251. Thanks.  

  1. Line 236-237: Pleas define βand  δ in formulas 11-12 

Response: Thanks. We added an explanation on Lines 272-273 (highlighted).

  1. Section 4.1: Please give the process used for calculating Weighting factors

Response: OK. We added a formula and the explanation on Lines 223-226 (highlighted) and hope it to be clearer.

  1. Discussion section should critically discuss the findings/results of the case study in conjunction to findings of previously published case studies.

Response: Thanks. For clarity, we added a statement to indicate that till the recent ten years (from 2009 to 2019), coordinated development of the 13 cities covering intensive development levels is less to study. Indeed, the modelling and the findings intends to be as the reference/research-basic of future development or other similar cites, especially considering the intensive coverage on social, economic and environment aspects. It is also highlighted on Lines 53-55.

Thanks for your reviewing effort !

Reviewer 2 Report

This paper uses 13 cities as a case study to characterize and quantify the relationship between economic, social, and environmental benefits from public transportation infrastructure investments.  The findings can be useful to transportation planners, if used in context with their own place-based analysis of their city and their sustainability goals.

Abstract could use a sentence that clarifies the main conclusion – what did you find? Are you recommending other urban areas use your model? Who should be using your work?  The last few paragraphs of the paper contain these strong conclusions, but the paper could be stronger if these were highlighted up-front in the abstract.

Whole paper could use a grammar check, there are minor grammatical errors that make sentences difficult to understand.  Should be easily fixed.

Line 40: I am unclear what you mean by geographical environment. Every city has geography, that is not unique.

Lines 45-48: This sentence is good, but it does not explain what the results of the model are useful for (or who would use the results). You should explain this very early so the work does not appear to be just a case study.

Line 55-58: this sentence is confusing. Recommend remove the word “even” and then change the wording to be clear about advantages or disadvantages.  I do not understand how a social benefit is a disadvantage.  I think maybe you are trying to write that investment in PT infrastructure strictly aimed at economic benefits is overlooking the other dimensions of the development, namely social impacts and environmental impacts.

Line 64: grammar issue, recommend removing the words “and what”

Line 75: In Figure 1, Student is spelled wrong in index U24.

Line 86: grammar issue, recommend removing the words “concerned with”

Line 126: grammar error, recommend adding “be” as in “is said to be” or simply remove “said to”

Lines 134-149: the generalized statements in this paragraph could be easily refuted by skeptics who may believe some of the opposite relationships exist (e.g. more passengers on PT causes more carbon emissions).  Recommend including a citation here and there to back up your summary.  Possibly some references in the vein of international development and the links to transportation infrastructure? Maybe something related to UN Sustainable Development Goals around transportation (related to goals 3, 9, and 11)?  https://sustainabledevelopment.un.org/topics/sustainabletransport

Line 158: recommend defining acronyms at first use.  GDP is not defined anywhere in the paper.

Line 184: You should be more clear on which value you mean when you say “when the indicator value is negative.”  Which indicator? Yij, or the growth rate (X) is negative?

Line 218: I think you repeated a sentence from line 203.  Equations 9 and 10 do not have Uli and p in them so there is no need to define those variables again here.

Line 225: In Table 2, you might want to explain how you decided on the splits between the values for H, and why you have the number of significant digits on the calculation.

Line 232: Grammar error, recommend rephrasing to “Because the sample set contains data…”

Line 236: there may be an error in equation (11), should the U variables all be U1 or should they be U1, U2, and U3?

Lines 237-242: Recommend you explain what the constant term (lambda-not) is, and ensure you define all variables (including the delta symbol).

Lines 253-268: Figure 2, all the graphs could benefit from defining U1, U2, and U3 in the legend.  There is plenty of room on each graph to have “U1: economic” and “U2: Social” etc., and even “S: Infrastructure Benefits” and “H: Coupling value.”  Also, to make the figures easier to read in greyscale printings, you should pick different symbols for each data series. Maybe asterisk, dots, squares, triangles, etc.

Lines 287-293: Your discussion here is very interesting.  You need to think deeper about the difference in urban setting for these cities.  Urban cities rely on PT in very different ways than rural cities.  Could there be an effect emerging here of “poor” cities vs. “More developed” cities just having different needs from transportation in general?  The types of jobs, the economic base for the city, the requirements that drive transportation in the first place can be different depending on the behaviors in that city.  It is worth at least acknowledging that these data must be used in context with the cities they represent.

Line 323: recommend you clarify the statement “air quality rate has risen” to specify whether air quality rising is good or air quality rising is bad (as in higher particulates, higher NOx, higher SOx).  Maybe use the word “improved” instead of “risen” if that is what you mean.

Line 329: your mention of increased private car travel begs the question about ridership on the metro lines. If the metro lines are at capacity, the rise in car travel may be a function of PT’s inability to meet the demand.

Section 4.2 could benefit from a general analysis of the temporal aspects of your results. Are there specific reasons for the spikes and dips in certain years, and is this normal due to investment in construction and then a gap before more investment happens?  And, do these investments include maintenance spending or is this just new construction?

Line 408: It looks like you do not have a single asterisk in either Table 6 or Table 7, only double-asterisks. So, maybe you only need half of this note?

Line 421: recommend you clarify what you mean by “during a low level.”  A low level of what?

Lines 425-428: This is a very important conclusion from your work. This sentence, or some version of it combined with line 430, belongs in the abstract. It gives the “so what” for your research.

Recommend adding to the abstract a condensed version of lines 441-445 as well.

References:  There may be typographical errors or missing information in references 6, 8, and 35.

Author Response

Response to reviewers and the editor- ijerph-911535, Sep.-2020

Coordination investigation of the economic, social, and environmental benefits of urban public transport infrastructure in 13 cities, Jiangsu Province, China

Xinghong He, Zhichao Cao, Silin Zhang*, Shumin Liang, Yuyang Zhang, Tianbo Ji, and Quan Shi

Note: All changes and revisions made in the revised manuscript are highlighted for convenience.

Reviewer: 2

Comments: This paper uses 13 cities as a case study to characterize and quantify the relationship between economic, social, and environmental benefits from public transportation infrastructure investments. The findings can be useful to transportation planners, if used in context with their own place-based analysis of their city and their sustainability goals.

Response: Thanks. We appreciate your view much.

  1. Abstract could use a sentence that clarifies the main conclusion - what did you find? Are you recommending other urban areas use your model? Who should be using your work? The last few paragraphs of the paper contain these strong conclusions, but the paper could be stronger if these were highlighted up-front in the abstract.

Response: Thanks for your advice. We added the clarification at the end of Abstract (highlighted). Meanwhile, we also explained the model proposed can be recommended as the fundamental method and various examples to refer.

  1. Whole paper could use a grammar check, there are minor grammatical errors that make sentences difficult to understand. Should be easily fixed.

Response: Thanks for your kindly reminding. We also asked for a professional editor to check/revise the whole paper and make sure the language.

  1. Line 40: I am unclear what you mean by geographical environment. Every city has 

geography, that is not unique.

Response: Good spotting. Thanks. A related explanation was added on Lines 46-47 (highlighted) to make it clearer.

  1. Lines 45-48: This sentence is good, but it does not explain what the results of the model are useful for (or who would use the results). You should explain this very early so the work does not appear to be just a case study.

Response: Following your comment, we added the explanation to indicate the availability by using our results (lines 54-57), highlighted. Thanks.

  1. Line 55-58: this sentence is confusing. Recommend remove the word "even" and then change the wording to be clear about advantages or disadvantages. I do not understand how a social benefit is a disadvantage. I think maybe you are trying to write that investment in PT infrastructure strictly aimed at economic benefits is overlooking the other dimensions of the development, namely social impacts and environmental impacts.

Response: Thanks. Indeed we accepted your suggestion and thought your expression to be clearer. Hence, we replaced with highlight on Lines 62-64.

  1. Line 64: grammar issue, recommend removing the words “and what".

Response: Thanks. Removed already on Lines 70-71.

  1. Line 75: In Figure 1, Student is spelled wrong in index U24.

Response: Good spotting and corrected in Figure 1. Thanks.  

  1. Line 86: grammar issue, recommend removing the words "concerned with".

Response: OK and removed. Thanks.

  1. Line 126: grammar error, recommend adding "be" as in "is said to be" or simply remove "said to".

Response: Changed on Line 150. Thanks.

  1. Lines 134-149; the generalized statements in this paragraph could be easily refuted by skeptics who may believe some of the opposite relationships exist (e.g. more passengers on PT causes more carbon emissions). Recommend including a citation here and there to back up your summary. Possibly some references in the vein of international development and the links to transportation infrastructure? Maybe something related to UN Sustainable Development Goals around transportation (related to goals3,9,and11)? https://sustainabledevelopment.un.org/topics/sustainabletransport

Response: Thanks for your professional suggestion. We added the necessary references and the link to make the view stronger in Section 3.1.

  1. Line 158:recommend defining acronyms at first use. GDP is not defined anywhere in

 the paper.

Response: We added the explanation of GDP (highlighted) behind Table 1.

  1. Line 184:You should be more clear on which value you mean when you say "when 

the indicator value is negative." Which indicator? Yij, or the growth rate (X) is negative?

Response: Thanks. We added the explanation to make it clearer. It is also highlighted on lines 208-211.

  1. Line 218: I think you repeated a sentence from line 203. Equations 9 and 10 do not have Uli and p in them so there is no need to define those variables again here.

Response: Good spotting again! We have checked and removed the duplicate definition.

  1. Line 225: In Table 2, you might want to explain how you decided on the splits between the values for H , and why you have the number of significant digits on the calculation.

Response: Thanks. For clarity, we added an explanation on Lines 250-251(highlighted). That is, the number of significant digits is derived from the acceptable criteria [40].

16.Line 232: Grammar error, recommend rephrasing to “Because the sample set contains data.…."

Response: Following your advice, changed on Lines 260-261. It is also highlighted. Thanks.

17.Line 236: there may be an error in equation (11), should the U variables all be U1 or should they be U1,U2,and U3?

Response: Thank you for pointing that out. We added the explanation on Lines 268-270.

18.Lines 237-242:Recommend you explain what the constant term (lambda-not) is, and ensure you define all variables (including the delta symbol).

Response: Thanks for pinpointing it. The definition of “constant term” is proposed, highlighted on Lines 270-273. Meanwhile, we checked all variables and ensured them.

19.Lines 253-268:Figure 2, all the graphs could benefit from defining U1,U2, and U3 in the legend. There is plenty of room on each graph to have "U1: economic"”and "U2:Social" etc., and even“S: Infrastructure Benefits" and "H:Coupling value.”Also, to make the figures easier to read in greyscale printings,you should pick different symbols for each data series. Maybe asterisk ,dots ,squares ,triangles , etc.

Response: Thanks. Following your good suggestion, we revised all sub-figure with different symbols for each data series. Meanwhile, we re-state the legends of Figure 2 with highlight on Line 282 to 284.

20.Lines 287-293:Your discussion here is very interesting. You need to think deeper about the difference in urban setting for these cities. Urban cities rely on PT in very different ways than rural cities. Could there be an effect emerging here of "poor” cities vs. "More developed" cities just having different needs from transportation in general? The types of jobs the economic base for the city, the requirements that drive transportation in the first place can be different depending on the behaviors in that city. It is worth at least acknowledging that these data must be used in context with the cities they represent.

Response: Thanks for your professional comments. We added explanation highlighted on Lines 325-329.

  1. Line 323: recommend you clarify the statement "air quality rate has risen" to specify whether air quality rising is good or air quality rising is bad (as in higher particulates, higher NOx, higher SOx).Maybe use the word "improved" instead of "risen” if that is what you mean.

Response: Yes, you are right. We changed on Line 359 (highlighted).

  1. Line 329: your mention of increased private car travel begs the question about ridership on the metro lines. If the metro lines are at capacity, the rise in car travel may be a function of PT's inability to meet the demand.

Response: Thanks for your opinion. We agree with your view: indeed the construction and operation of metro have multiply effects. Hence, we cannot assess metro development to improve or damage the environmental benefits and decided to move unclear “metro issue” here. In addition, the capacity of PT or metro is much more than private transport in general, so that the environmental benefits of PT is also better by the average comparison.

  1. Section 4.2 could benefit from a general analysis of the temporal aspects of your results. Are there specific reasons for the spikes and dips in certain years, and is this normal due to investment in construction and then a gap before more investment happens? And, do these investments include maintenance spending or is this just new construction?

Response: Thanks for your question. We added the explanation with highlight on Lines 403-407.

  1. Line 408: It looks like you do not have a single asterisk in either Table 6 or Table 7, only double-asterisks. So, maybe you only need half of this note?

Response: Thanks again. We changed on Line 449.

  1. Line 421: recommend you clarify what you mean by “during a low level.”A low level of what?

Response: Thanks again. We changed on Lines 461-463.

  1. Lines 425-428: This is a very important conclusion from your work. This sentence, or some version of it combined with line 430, belongs in the abstract. It gives the “so what ” for your research.

Response: Thanks. We changed on Lines 33-34. Highlighted.

  1. Recommend adding to the abstract a condensed version of lines 441-445 as well.

Response: Thanks. Similar to the above revision, we added the sentence by following your advice. It is also highlighted on Lines 33-34.

  1. References:There may be typographical errors or missing information in references 6, 8, and 35.

Response: Thanks. Corrected already.

Thanks for your reviewing effort !

Round 2

Reviewer 1 Report

The paper seems strongly improved. Most of my concerns from my previous review have been addressed.